# Establishment of a Sheep Model for Hind Limb Peripheral Nerve Injury: Common Peroneal Nerve

**DOI:** 10.3390/ijms22031401

**Published:** 2021-01-30

**Authors:** Rui D. Alvites, Mariana V. Branquinho, Ana C. Sousa, Federica Zen, Monica Maurina, Stefania Raimondo, Carla Mendonça, Luís Atayde, Stefano Geuna, Artur S.P. Varejão, Ana C. Maurício

**Affiliations:** 1Departamento de Clínicas Veterinárias, Instituto de Ciências Biomédicas de Abel Salazar (ICBAS), Universidade do Porto (UP), Rua de Jorge Viterbo Ferreira, nº 228, 4050-313 Porto, Portugal; ruialvites@hotmail.com (R.D.A.); m.esteves.vieira@gmail.com (M.V.B.); anacatarinasoaressousa@hotmail.com (A.C.S.); cmmendonca@icbas.up.pt (C.M.); ataydelm@gmail.com (L.A.); 2Centro de Estudos de Ciência Animal (CECA), Instituto de Ciências, Tecnologias e Agroambiente da Universidade do Porto (ICETA), Rua D. Manuel II, Apartado 55142, 4051-401 Porto, Portugal; 3Department of Clinical and Biological Sciences, Cavalieri Ottolenghi Neuroscience Institute, University of Turin, Regione Gonzole 10, 10043 Orbassano, TO, Italy; federica.zen@unito.it (F.Z.); monica.maurina@edu.unito.it (M.M.); stefania.raimondo@unito.it (S.R.); stefano.geuna@unito.it (S.G.); 4CECAV, Centro de Ciência Animal e Veterinária, Universidade de Trás-os-Montes e Alto Douro (UTAD), Quinta de Prados, 5001-801 Vila Real, Portugal; avarejao@utad.pt; 5Departamento de Ciências Veterinárias, Universidade de Trás-os-Montes e Alto Douro (UTAD), Quinta de Prados, 5001-801 Vila Real, Portugal

**Keywords:** peripheral nerve injury, peripheral nerve regeneration, common peroneal nerve, animal model, sheep model, nerve anatomy, neurological exam, nerve stereology

## Abstract

Thousands of people worldwide suffer from peripheral nerve injuries and must deal daily with the resulting physiological and functional deficits. Recent advances in this field are still insufficient to guarantee adequate outcomes, and the development of new and compelling therapeutic options require the use of valid preclinical models that effectively replicate the characteristics and challenges associated with these injuries in humans. In this study, we established a sheep model for common peroneal nerve injuries that can be applied in preclinical research with the advantages associated with the use of large animal models. The anatomy of the common peroneal nerve and topographically related nerves, the functional consequences of its injury and a neurological examination directed at this nerve have been described. Furthermore, the surgical protocol for accessing the common peroneal nerve, the induction of different types of nerve damage and the application of possible therapeutic options were described. Finally, a preliminary morphological and stereological study was carried out to establish control values for the healthy common peroneal nerves regarding this animal model and to identify preliminary differences between therapeutic methods. This study allowed to define the described lateral incision as the best to access the common peroneal nerve, besides establishing 12 and 24 weeks as the minimum periods to study lesions of axonotmesis and neurotmesis, respectively, in this specie. The post-mortem evaluation of the harvested nerves allowed to register stereological values for healthy common peroneal nerves to be used as controls in future studies, and to establish preliminary values associated with the therapeutic performance of the different applied options, although limited by a small sample size, thus requiring further validation studies. Finally, this study demonstrated that the sheep is a valid model of peripheral nerve injury to be used in pre-clinical and translational works and to evaluate the efficacy and safety of nerve injury therapeutic options before its clinical application in humans and veterinary patients.

## 1. Introduction

Peripheral nerve injuries (PNIs) affect a large number of people every year, and the number is likely to increase [1,2]. In addition to the functional and psychological consequences directly related to the injuries, PNIs are also associated with high socioeconomic impacts and costs worldwide [3]. There are several causes of PNIs, mostly associated with accidents, penetrating trauma, fire gun injuries, crushing and stretching after falls, lacerations after neighboring bone fractures, and injuries of iatrogenic origin [4]. 

PNI classification is still done according to Seddon [5] and Sunderland [6], despite some updates over time. Within this classification, axonotmesis and neurotmesis are the two most common experimental lesion paradigms to be explored. Regarding axonotmesis, an axonal lesion occurs, but most connective tissue layers involving the nerve (endoneurium, perineurium, and epineurium) are wholly or partially intact, allowing nerve regeneration to occur. This situation usually arises as a result of crushing or stretching injuries [7,8]. Neurotmesis is the most serious type of lesion, with injury of the axons, layers of connective tissue and myelin sheaths. Functional recovery is always suboptimal and surgical intervention is essential to ensure recovery and reinnervation. It usually originates from severe trauma, penetrating and destructive injuries, avulsion and traction lesions and local injection of harmful substances [8,9].

The most serious PNI lesions, when complete transection occurs, require surgical intervention to reconnect the proximal and distal nerve tops through anastomosis, the use of natural or synthetic grafts or biomaterials and nerve guidance conduits (NGCs) [10]. In some cases, axonotmesis lesions also benefit from wrapping with nerve conduits to accelerate nerve regeneration [7]. However, the prognosis of surgical nerve repair to solve PNI remains suboptimal, regardless of the site of injury or the chosen therapeutic technique [11]. 

Poor nerve regeneration quality is generally related to long distances between nerve tops and low rates of regeneration, leading to prolonged denervation periods. This factor negatively affects the ability of distal nerve structures to support regeneration and reinnervation of target organs [12]. The creation and development of new approaches to stimulate peripheral nerve regeneration, although necessary, require the use of reliable preclinical models that adequately mimic the challenges associated with clinical PNI [13]. 

Traditionally, peripheral nerve regeneration studies have been conducted in small animal models such as rodents, but the differences between species are several and limit the clinical translation of the obtained results. The injury dimensions considered as long gap nerve defects in small animals are much smaller, even proportionally, than the critical nerve gaps considered in humans [14]. Moreover, the proportion of connective tissue in rodent nerves is different, and the extremely high neuroregenerative rate of this species makes it difficult to gauge the true effectiveness of the therapies instituted in promoting nerve repair [7,15]. Larger animal models are ideal for simulating the long-distance nerve defects and regenerative phenomena that are observed in human PNI. Since these models can more reliably replicate some characteristics of human nerves, such as their structural composition, dimensions, diameter and regenerative process, there are already several published studies based on the use of dogs [16,17], cats [18], rabbits [19], non-human primates [20], pigs [13,21], mini-pigs [22], guinea-pigs [23], and sheep models [24].

The objective of the present study was to establish an alternative model of surgical injury in a sheep peripheral nerve, namely the common peroneal nerve in the hind limb. We opted for a model in the hind limb due to its importance in the dynamics of sheep’s gait and environmental exploration, mimicking the functional consequences observed in humans when lower limb injuries occur. The choice of the common peroneal nerve intended to avoid the severe and limiting functional consequences of animal welfare associated with an injury to the sciatic nerve, however allowing to observe evident motor and sensory impairments and monitor its progression over time. For this purpose, the two most common experimental lesion paradigms were induced, and different therapeutic techniques were applied to obtain preliminary results that allow nerve regeneration studies to be carried out in the future. A neurological examination protocol adapted and directed to the common peroneal nerve was stablished and used to assess the animal’s functional recovery over the study period, and the injured and treated nerves were collected for morphological and stereological evaluation. These last two assessments had the sole objective of understanding whether the applied injury models allow the subsequent performance of a functional and histomorphometric assessment, obtaining preliminary results as control values for future studies and stablishing a neurological exam easily replicable.

### 1.1. Sheep as an Animal Model of Peripheral Nerve Injury

The sheep is a large animal model with great relevance in studies of nerve regeneration prior to translation to humans [25]. Not only does the sheep have similar body and peripheral nerve dimensions to humans, but the rate of nerve regeneration in the two species is identical [26,27]. In addition, sheep nerves are histomorphologically identical to those of humans, being polyfasciculated [28]. In studies where the age of the animal model is an important factor, the sheep model is also advantageous since the age correspondence between this species and humans is well determined [29]. Recently, the sheep’s kinematic patterns have also started to be studied, creating new tools to be applied in this species in PNI models [30].

Additionally, sheep are docile-tempered animals, easy to obtain and maintain, and their use raises fewer ethical questions than with other species [31]. Despite all the advantages, the number of studies performed with sheep in the field of peripheral nerve regeneration is still limited. Most of the studies were performed at the median and facial nerves, with the hind limb nerves having little representation. In addition, most studies have focused on induction of neurotmesis lesions, and few studies have been done on axonotmesis [25,32,33]. Finally, there is not yet a widely accepted and easily reproducible sheep injury model to be used in an experimental environment.

### 1.2. Anatomy and Topographic Relationships of the Common Peroneal Nerve

Different nerves may have similar clinical presentations after injury, namely branching nerves. In the case of the present study, considering the branching of the sciatic nerve into tibial and common peroneal nerves, the injury of the main nerve will manifest through a combination of the clinical signs related to the injury of each individual branch. It is important to have a thorough understanding of the anatomical and topographic distribution of the common peroneal nerve and of the nerves that are anatomically related to it (Figure 1) as to comprehend their functions and regions of innervation, as well as the direct functional consequences associated with their injuries.

The lumbosacral plexus originates the nerves responsible for the hind limb innervation, excluding some proximal skin regions. The plexus consists essentially of the ventral branches of the last lumbar nerves and first sacral nerves [34,35]. After originating the obturator nerve and femoral nerve and its saphenous nerve branch [34,36], the remaining branches of the lumbosacral plexus originate from the common lumbosacral, which is essentially formed from the last lumbar nerves and the first two sacral nerves.

The sciatic nerve is the distal continuation of the lumbosacral trunk. The nerve leaves the pelvic cavity, advancing through the dorsal and caudal regions of the hip, innervating the caudal muscles of the thigh and being protected by the great trochanter of the femur. Before reaching the gastrocnemius muscle, the nerve divides into the tibial and common peroneal nerves, which together ensure the innervation of all structures distal to the stifle, except for the medial region innervated by the saphenous nerve [34,36,37,38].

The tibial nerve branches from the sciatic nerve at the proximal level of the gastrocnemius muscle, and more distally divides into the medial and lateral plantar nerves. The plantar nerves innervate and guarantee the sensation of the plantar region of the foot [34,37,38].

The common peroneal (fibular) branches from the sciatic nerve after its emergence from the pelvic cavity, crossing the gastrocnemius medially to the biceps muscle and becoming superficial behind the lateral collateral ligament of the stifle joint. Distally, the nerve becomes deep again, advancing between the peroneus longus and lateral digital extensor muscles before detaching the lateral sural nerve (skin innervation of the hind limb lateral aspect) and dividing into the superficial and deep branches near the head of the fibula. The larger superficial peroneal nerve crosses the long peroneal muscle in a deep position before advancing to the foot, innervating the skin of the dorsal region of the leg and foot. The deep peroneal nerve, on the other hand, innervates the dorsolateral muscles of the leg, namely the extensor muscles of the digits and ensures sensory innervation of dorsal region of the foot [34,37,38].

### 1.3. Functional Consequences of Injuries Directed to Common Peroneal and Topographically Related Nerves

Paralysis of the common peroneal nerve usually originates from direct trauma to the lateral stifle region where the nerve is superficial and is evidenced by overextension of the hock and overflexion of the distal joints, in addition to severe sensory deficits and loss of skin sensation in the cranial-dorsal region of the metatarsus and digits. The limb will rest on the dorsal surface of the flexed digits unless properly positioned passively. Eventually the animal learns to compensate for this change by flicking the foot forward before placing the plantar surface in the ground [34,36,39]. Due to the close topographic relationships with the tibial nerve, and both being branches of the sciatic nerve, the functional consequences of the injury to these three nerves can overlap and it is important to distinguish the changes resulting from the lesion of each one. 

Sciatic nerve severe injuries result in a suspended and loose limb, extension of the stifle and hock joints, flexion of the digital joints and the foot is knuckled. The patellar reflex will be normal or increased since the quadriceps contraction reflex coordinated by the femoral nerve will not be opposed by the muscles innervated by the sciatic nerve. The cutaneous sensation of the extremity is lost. Symptomatology is basically the combination of manifestations of lesions of the tibial and common peroneal nerves, but the fixation of the stifle joint through the quadriceps, if the femoral nerve is intact, can allow the support of some weight [34,36,37]. 

Tibial nerve damage manifests through a hock overflexion and a fetlock overextension, the pastern displaying a vertical position. Since the digital extensor muscles are unaffected, hooves are positioned normally and the animal can walk, and can support its weight while at rest. Despite this, anomalous joint dynamics are observed during gait [34,36].

### 1.4. Neurological Exam of the Common Peroneal Nerve

Identifying the clinical manifestations associated with common peroneal nerve injury is essential to confirm the lesion of the intended nerve and the recovery over time. The primary goal of neurological examination is to determine if the nerve is affected. Repeating the neurological exam over time will allow to determine improvements. The neurological examination should include an assessment of the animal’s mental status and free movement capacity, posture, postural reactions, and nociceptive response [40].

Determining the animal’s mental status before initiating a hind limb approach is important to ensure that the sheep is alert, interacts normally with the environment, is aware of the examiner’s presence, maintains its gregarious behavior (when housed with other animals) and is capable of free movement (regardless of deficits associated with nerve damage). Changes in this state of normality may hamper the animal’s response in the following stages. The posture of the animal, when in stationary position, must be evaluated, namely, to determine the normal positioning of the limb in relation to the body axis and to the ground. Once the animal is determined to be able to walk safely, it must do it in a closed but wide area with forward and backward movements. Hind limb movements should be analyzed, and attention should be paid to foot positioning when the animal changes direction or gain speed, determining whether the gait is normal, symmetrical and consistent, whether the posture is normal or ataxia is observed, if the injured limb is affected, abducting and waning in changes of direction, and also whether the limb interferes or knuckle during gait [36,40].

Assessment of postural reactions is particularly useful for identifying asymmetries between the injured and healthy limbs. Proprioceptive positioning allows to teste the proprioceptive integrity, to be assessed by placing the animal’s limb in an abnormal position, which should be corrected immediately (Figure 2a). The test can be difficult to perform on nervous, aggressive and uncooperative animals, and since sheep do not easily tolerate their limbs being manipulated, results are not always easy to interpret. A good alternative is a dynamic positioning test, in which the animal’s limb is placed on a mobile platform (plastic, piece of wood), the platform being moved slowly away from the animal, determining the time it takes to reset the member in its original position (Figure 2b) [41]. 

Spinal reflexes should be tested with the animal in a lateral recumbency, with the limb to be evaluated facing upwards. The withdrawal reflex of the hind limb evaluates the sensory and motor component of the sciatic nerve and its ramifications, eliciting flexion at all joints of that limb. The reflex is stimulated by pinching the lateral digit, observing the flexion of the limb when there is integrity of the common peroneal nerve (Figure 2c). The skin over the medial digit should also be pinched to ensure femoral nerve integrity and flexion of the hip. Pinching the plantar region of the foot will trigger the same reflex by stimulating the terminal branches of the tibial nerve. In a normal withdrawal reflex, the animal should move the limb away from the painful stimulus by showing awareness through vocalization or by looking at the tested limb. If the animal does not respond initially, an increase in stimulus intensity should be tested before considering the reflex to be absent [36,40,41].

Pain may be difficult to assess in sheep as they are tendentially stoic animals. Once there is no specific method for determining the presence of pain, signs should be observed during the remaining neurological examination, indicating the presence of pain perception.

## 2. Results

### 2.1. Animals

The choice of the sheep as an animal model proved to be advantageous, with the animals showing a personality and behavior that facilitated its manipulation and interaction throughout the work, namely during the neurological examinations. The applied anesthetic protocols allowed a smooth surgical induction, maintenance and recovery without unexpected hurdles, with analgesia and antibiotics guaranteeing the animals’ well-being in the post-surgical period and the absence of unexpected complications.

### 2.2. Surgery

The local block performed on the common peroneal nerve allowed the surgical intervention to be carried out without any manifestation of pain perception by the animal. The technique of surgical access with exposure of the common peroneal nerve through a lateral incision along the thigh proved to be largely advantageous, with a quick, simple and wide access to the nerve that allowed an easy induction of surgical injuries and subsequent application of the therapeutical approaches. 

### 2.3. Nerve Injury and Therapeutic Options

As with lesion induction, the application of different therapeutic approaches was easily achieved. Both neurotmesis and axonotmesis injuries were easily induced, with enough space to manipulate the nerve and surgical material without causing excessive nerve stretching or injury to neighboring tissues. The dimensions of the nerve facilitated the performance of the crush injury and the application of sutures in the EtE treatments and in the use of NGCs, with the diameter of the nerve adapting perfectly to the internal dimensions of the applied NGC. 

### 2.4. Neurological Evaluation

Immediately after the animal’s surgical recovery, all animals, regardless of the type of injury induced, showed clinical signs indicative of common peroneal nerve damage. The collection of data from the neurological examination over the study period allowed the confirmation of post-injury neurological deficits, as well as progressive clinical improvements over time. 

#### 2.4.1. Mental Status

All animals showed no changes in mental status over the study period, and no behavioral deviations were identified that could compromise the results obtained during neurological examinations, neither in the perisurgical period nor in the subsequent phases.

#### 2.4.2. Posture

After the induced injuries, all animals showed severe postural changes, that is, overextension of the hock and overflexion of the distal joints, with the limb resting on the dorsal surface of the flexed digits with both the animal standing and moving (Figure 3). Some animals showed compensatory behavior, flicking the foot forward to place the plantar surface in the ground. No clinical improvements were observed until week 4 for all the therapeutic groups. In the fourth week the axonotmesis group began to show postural improvements. In the following weeks, the remaining groups began to show postural improvements with continuous progression until the end of the respective study periods. After 12 weeks, the axonotmesis group showed complete postural recovery. At 24 weeks the group that received NGCs showed better results than the animals where EtE sutures were applied (Figure 4).

#### 2.4.3. Free Movements

All animals maintained their normal ability to perform free and voluntary movements over the study time, and without manifestation of discomfort and/or pain even with postural changes associated with surgical intervention.

#### 2.4.4. Proprioceptive Assessment: Static Repositioning

In the first week after the injury, all animals subjected to neurotmesis showed deficits of proprioception, with an inability to restore the limb to its physiological position after the forced placement of the dorsal surface of the digits against the ground within the expected physiological period (or taking more than 3 s to reposition). Animals with axonotmesis lesions showed better proprioceptive repositioning times in the first week, performing repositioning in less than 5–10 s. In animals with neurotmesis, the groups treated with EtE sutures began to show improvements in proprioception in the second week after surgery and the groups that received NGCs only after the eighth week. At the end of the study period, the animals in the axonotmesis group presented proprioceptive repositioning times of 3–5 s. At 24 weeks, both animals in the EtE group and those that received NGCs showed proprioceptive repositioning times of 5–10 s. Despite this, none of the groups reached proprioceptive repositioning values below the 3 s observed in the non-injured limbs (Figure 5). 

#### 2.4.5. Proprioceptive Assessment: Dynamic Repositioning

Like in the static repositioning, in the first week after the injury, all animals subjected to neurotmesis showed proprioception deficits, with an inability to restore the limb to its physiological position within the expected physiological period after it was placed on a moving platform and laterally moved away from the body axis, with a slow and continuous movement (or taking more than 3 s to reposition). Animals with axonotmesis lesions showed better times of proprioceptive repositioning in the first week, with animals performing repositioning in less than 5–10 s. In animals with neurotmesis, the groups treated with EtE sutures showed improvements in proprioception in the sixth week and the groups that received NGCs only after the eighth week. At 12 weeks, the animals subject to axonotmesis showed times of proprioceptive repositioning of 3–5 s, as the group treated with NGCs at the end of 24 weeks. At the same timepoint, animals treated with EtE suture took 5–10 s to perform the proprioceptive repositioning. Despite this, none of the groups reached proprioceptive repositioning values below the 3 s observed in the non-injured limbs (Figure 6). 

#### 2.4.6. Withdrawal Reflex

In the first week after surgical injuries, all therapeutic groups showed absence of withdrawal reflex when pinched in the lateral digits (skin and hooves). In the groups that received a crush injury, the return of the reflex was observed after the fourth week. On the other hand, in groups with neurotmesis, the reflex reappeared from the sixth week on animals that received EtE sutures and from the eighth week on animals that received NGCs. At the end of the study period, animals from axonotmesis group and neurotmesis group treated with NGCs and EtE sutures showed a standard reflex with the tested limb being moved away from the painful stimulus and the animal manifesting awareness of the received stimulus (Table 1). 

### 2.5. Nerve Morphological and Stereological Analysis

At the time of collection, differences in the presentation of the nerves were observed. In neurotmesis, the animals in the groups that received NGCs showed nerves involved by a significant amount of fibrous tissue and adhered to the neighboring tissues, making it difficult to individualize and to harvest (Figure 7a). The nerves of animals submitted to EtE showed lower levels of fibrous adhesions and the harvesting was easier (Figure 7b), as well as in the nerves of animals subjected to axonotmesis. All efforts were made to atraumatically isolate the nerves and the injury site, maintaining their integrity. The stereological results obtained after analysis of healthy nerves can be consulted in Table 2.

Since healthy nerves presented fibers with big differences in size, it is important to present also the frequency distribution of fiber diameter (Figure 8), which can help to appreciate changes during regeneration in the amount of small and big fibers in future works. Regarding the distribution of the fiber diameter in the healthy nerve, it was possible to perceive that there was a wide distribution, and although more than 70% of the fibers have a diameter of less than 11 µm, larger myelinated fibers were also observed.

The images of toluidine blue stained sections representative of all the groups considered can be seen in Figure 9 and Figure 10. Images representing healthy nerves (Figure 6a and Figure 7a) confirm data about the wide distribution of fibers size, presented in Figure 5. 

Both healthy (Figure 9a and Figure 10a) and regenerating nerves (Figure 9b–d and Figure 10b,c) showed, at low magnification (first column of Figure 9 and Figure 10) many fascicles. All nerves subject to injury and/or therapeutic intervention showed regenerating fibers already at 12 weeks after surgery, surprisingly also in the neurotmesis group repaired with the conduit (Figure 9d) where the regeneration seems to be better than in the neurotmesis + EtE group (Figure 9c). At 24 weeks, EtE (Figure 10b) and NGC (Figure 10c) groups present a more complete regeneration than after 12 weeks. EtE group showed a higher density of fibers with a thicker myelin than the NGC group. In comparison to the healthy nerves, the regenerating nerves of all groups (12 and 24 weeks) showed microfasciculation inside each fascicle, characteristic that corroborates the occurrence of a regenerative process. Moreover, regenerating myelinated fibers are visibly, smaller and with a thinner myelin sheath than in healthy nerves. The evaluation of nerves subjected to axonotmesis revealed the presence of healthy fascicles with no evidence of injury, even if in the presence of a small amount of degenerated and regenerated ones, indicating that the attempted crush injury was not complete (Data not shown). 

## 3. Discussion

In this study, we described a model for injury of sheep common peroneal nerve that allows one to mimic different scenarios of relevant nerve injuries in humans. Furthermore, a neurological exam protocol to monitor the animals’ functional recovery throughout the study periods was also adopted. For this, the animals were subjected to the most common experimental lesion paradigms, axonotmesis and neurotmesis, with the transected nerves receiving different types of therapeutic options, namely EtE sutures and NGCs, considering different study times. To test the feasibility of the established neurological exam, the animals were regularly monitored throughout the study periods to determine variations and functional improvements in the different parameters considered. At the end of the study period, the animals were euthanized and the intervened and healthy contralateral nerves were collected for morphological and stereological studies that allowed not only to establish the standard values for healthy sheep nerves (useful as controls in future studies) but also to identify preliminary histomorphological differences between study groups. The established model allows to open doors for future works comparing the therapeutic efficacy of different medical and surgical options associated with the most common injury paradigms, but it can also be adapted for other neurosurgical procedures such as the use of nerve transfers for reinnervation of injured nerves in other body segments, electrical stimulation to promote nerve regeneration and even the use of different biomaterials and cell-based therapies.

Most studies on nerve regeneration are based on the use of small animal models to replicate complex nerve injury scenarios [42]. However, these models do not perfectly mimic the biological complexity and regenerative processes observed in humans and other mammals so the translationality of the results is limited. Only large models are able to replicate the effects of injuries and the regeneration of larger nerve defects, which are those that continue to create real challenges in nerve regeneration and functional restoration in humans [43]. Although the choice of larger models is essential, the selection of the species requires some considerations. The selected species must be cost-effective, easy to manipulate, have a behavior easy to evaluate and with tissues that can be easily analyzed. Unlike other explored models, sheep have all the characteristics mentioned and their nerves are remarkably identical to those of humans, both in dimensions and in constitution [44]. Moreover, the well-known physiology of sheep has increased the popularity of the use of this species as a pre-clinical model for several medical fields such as spinal cord injury [45], traumatic brain injury [46,47], wound healing [48], bone regeneration [49,50], chondral diseases [51] and vascular disorders [52]. In fact, the objective of establishing the model presented in this work was to allow the replication of critical defects that are highly challenging from the point of view of repair and regeneration, simulating situations of severe PNI in humans. The choice of sheep as a large model proved to be advantageous. The selected breed allowed us to acquire cheap animals, easy to maintain and feed, relatively calm, easy to train and with predictable behaviors. Additionally, their dimensions also facilitated the performance of the neurological exam, namely in the observation of posture and free movements and its restraint during the proprioceptive and spinal reflexes assessments. Finally, the tendentiously developed musculature of the sheep hind limbs and the dimensions of the nerves, identical to those in humans, allowed the surgical access, nerve isolation, lesion induction and therapeutic interventions to be carried out effectively and with minimal trauma, allowing to stablish the surgical access as the ideal to be used in future works. In the second technique initially considered, with the exposure of the common peroneal nerve more distally at the location where the nerve branches, the peroneus longus and lateral digital extensor muscles should be identified, and the incision made between them. Subcutaneous debridement should be carried out in such a way as to separate the two muscles and to individualize the common peroneal nerve from neighboring tissues. In this case, while also allowing quick access, muscle development and the location close to the nerve branching site would create some difficulty in separating the peroneus longus and lateral digital extensor muscles for exposure and isolation of the nerve, as well as for the application of NGCs, leading to greater trauma, hemorrhage, and tissue reaction, which would result in difficulties to collect the nerve and to perform further stereological and histomorphological analysis. Additionally, considering the decrease in the diameter of the nerve in more distal positions, the internal diameter of the NGC used would be too large to allow a correct accommodation and suture of the nerve inside it and would make it difficult to accommodate the NGC between the muscles. The third initially considered surgical technique was also promptly disregarded due to the technical difficulties associated with its performance, even considering that this technique has already been used in other works [53]: Not only would the caudal access require great mobilization and trauma for the separation of the semitendinosus and semimembranosus muscles, but the exposure of the nerve would be minimal, hampering the performance of the nerve damage through the surgical window between the musculature.

The common peroneal nerve is the most commonly injured in humans lower limb when the peroneal division of the sciatic nerve is involved [54]. These injuries can have several origins, from penetrating and blunt trauma, chronic and acute compressions, diabetes, anesthesia, peripheral neuropathy and even idiopathic causes [55]. All of this justifies the choice of the corresponding nerve in the sheep model as the one to be studied and intervened. Although the anatomy and topographic distribution of this nerve in the sheep is not as complex as in humans, the lesions of the common peroneal nerve translate into identical symptoms in both species, humans showing plantarflexion and inversion of the foot with inability to dorsiflex the ankle [56] and the sheep expressing overextension of the hock and overflexion of the distal joints [40]. This symptomatic presentation is specific, easily identifiable and allows to confirm the correct induction of the lesion in the animal after surgery. Common peroneal nerve injuries have the advantage of not leading to such serious and disabling consequences as a sciatic nerve injury, allowing residual innervation of the hind limb and muscle groups that tolerates weight support even in the presence of functional and postural deficits. Additionally, an injury that affects the hind limb in quadriceps animals has no impact on the animal’s chest and head weight support, mitigating the traumatic effects for the animal, allowing it to express its normal feeding and exploration behavior and reducing the need for special requirements such as weight-bearing material. Deficits due to nerve damage are those that can be used to determine recovery over time, even if these measurements are time and resources consuming in the ovine model. Functional changes vary with time and compensatory mechanisms: while deficits are obvious during the acute phase of recovery, over time they become less evident even if they are still present. This fact, which can originate in a redundancy in the innervation or in mechanisms adopted by the animal to compensate for the injury (as for example flicking the foot forward before placing the plantar surface in the ground), can hinder the correct evaluation of the temporal recovery. 

The model presented in this study creates a platform to evaluate the therapeutic effects in the repair of nerve injuries with different degrees of severity (crushing and transection) over 12 and 24 weeks in a large model. The establishment of this model is important not only to mimic severe clinical situations in humans, but also in veterinary species such as the dog [57] or horse [58], where peripheral nerve injuries with different etiologies are relatively common. Observation of the characteristic symptoms of the common peroneal nerve injury after surgery shows that the nerve damage causes changes in both the motor and sensory branches of the nerve, which is expected due to it mixed nature [59]. This creates the need for indirect measurements tools to assess the severity of nerve damage and the progression of functional recovery during regeneration, namely through a neurological examination oriented to the hind limb and which includes evaluating the different components affected by the induced injury. The neurological exam protocol presented here results from the adaptation of general neurological examination protocols used in ruminants [36,40,41], which are generally an exercise of observation and are not as developed and established as those applied in companion animals and horses. Although it is time-consuming, the performance of this examination in the sheep allows to assess changes in posture, movement capacity, postural reactions and spinal reflexes resulting from nerve damage, as well as their progression over time. Due to the easy handling of these animals, two operators are sufficient to allow the performance of the tests that require restraint. Considering the small number of animals used in this work and the impossibility of determining statistical differences, neurological exams were not intended to determine the comparative effectiveness of therapeutic options in different types of lesions, but only to define whether the tests applied allowed to identify differences in functional recovery over time and to trace general results associated with regeneration to serve as a basis for future works. 

The preliminary results allowed us to observe that the different surgical interventions did not translate into changes in the animals’ mental status nor did interfere with the animal’s ability to move even with functional deficits. In animals subjected to neurotmesis, those that received NGCs showed earlier improvements in posture evaluation with the group that received EtE sutures expressing proprioceptive improvements and spinal reflexes earlier. However, at 24 weeks, the NGC group tends to have better final performances. The signs of functional recovery at earlier timepoints in the animals that received EtE may be related to the surgical technique itself, since the juxtaposition between the two nerve tops facilitates axonal reconnection and allows the nerve regeneration process to begin more quickly. However, perfect axon-to-axon or endoneurium-to-endoneurium alignment is always difficult to achieve, and the subsequent occurrence of aberrant motor/sensory connections in the regenerating nerve and even misdirection is common. Thus, in later timepoints, this group is functionally surpassed by the animals that received NGCs, what is in line with the various recent studies that indicate all the advantages of using nerve conduits compared to traditional surgical techniques [60], with the conduct to guarantee a pro-regenerative environment over time and a better general therapeutic performance [61]. Similarly, the results showed that a period of 12 weeks is insufficient to determine the real effectiveness of therapies instituted to promote nerve regeneration after neurotmesis in this model, and a period of 24 weeks should be used in further studies. This result is in accordance with the rate of peripheral nerve regeneration in larger animals, which is slower than that observed in rodents in which shorter study periods are considered [62].

In animals subjected to axonotmesis, there were also no changes in mental status and ability to move and walk, and the symptoms observed in the first week after injury seemed to demonstrate that the force applied by the clamp could have been sufficient to generate symptomatology compatible with crush injury, although the histomorphometric evaluation revealed absence of injury afterwards. According to Dahlin et al. [63], the degree of nerve damage in an axonotmesis injury is determined by two factors: the applied pressure and the crushing duration [64]. Initially, an axonotmesis protocol identical to that used in the rat model [65] was tested, in which the injury was induced by applying 54 N for 30 s creating a 3 mm long crush injury. Although after crushing the corresponding flattening of the nerve was observed and right after the surgery the animals showed compatible functional deficits, in the first neurological evaluation, a week later, the intervened animals did not present any identifiable deficit, which indicates that the induced injury was only a neuropraxia. The non-serrated clamp was then adapted to allow a force of 80 N, and a 5 mm long, with the force being applied for 1 min. In this case, and even if the derived pressure applied was not particularly high, longer crush times were used to compensate and trigger an effective injury, what initially appeared to have happened through the observation of the functional deficits. As expected, the functional recovery of all evaluated parameters started earlier in axonotmesis injuries comparing to neurotmesis injuries, and after 12 weeks of study, functional recovery was observed in almost all parameters.

Differences were detected that compromised the material collection. Compared to smaller models such as rodents, in the sheep model surgical intervention is always bloodier, and the degree of injury to neighboring tissues not only caused higher levels of bleeding during the surgical intervention, but also triggered higher levels of tissue adhesion and fibrosis, making it difficult to collect and correctly individualize the nerve after euthanasia. The fact that the common peroneal nerve has a superficial and more exposed location also increases the movements of the skin and surrounding tissues during gait, creating a more aggressive environment and conducive to the formation of tissue adhesions. Although fibrosis levels were present in all cases, they were particularly obvious in neurotmesis, more evident after the application of NGCs than after EtE sutures. The creation of a physical barrier between the pro-regenerative environment within the NGC and the aggressive neighboring environment is one of the advantages of using NGCs [66], and in this case it may justify the better performance of this therapeutic method compared to EtE, since in spite of the large amount of tissue surrounding the conduit, the regeneration within it took place effectively. In the case of axonotmesis induction, although some degree of fibrosis was also observed, it was evidently lower.

Stereology is a method of direct measurement which allows to quantify the characteristics of regenerated nerve fibers (namely their number and dimensions) and also the thickness of the myelin sheath surrounding the nerve by applying methods of quantitative and morphometric analysis in the histological sections under study. This quantitative analysis makes it possible to identify phenomena of inflammatory reactions, fibrosis, perineural adhesions, development of neuromas, quantify cells in certain regions of the nerve, determine the proportion between regenerated and injured tissue and, when using biomaterials, they also allow to determine the presence of foreign body reactions, granulomas and the level of material degradation [7,67]. However, for the stereological evaluation to be carried out, it is necessary that the nerves are collected following specific protocols and maintaining their structural integrity. In this work, the common peroneal nerves were collected after the established study periods, both for the intervened nerves and for healthy contralateral nerves. As mentioned before, the objective of this evaluation was to determine the stereological characteristics of the healthy nerves to establish control values to be used in future works, in addition to identifying general histomorphological characteristics of the nerves intervened and that received therapeutic options to be evaluated as preliminary and guiding results. 

The observation of some degenerated/regenerated fascicles in conjunction with healthy fascicles on the nerves subjected to axonotmesis indicates that the induced crush injury was not effective from the structural point of view, even if it has been translated into compatible symptoms in vivo. Of course, this incomplete injury also justifies the good functional performance observed in this group in the neurological evaluations performed, with complete functional recovery observed in almost all parameters considered after 12 weeks. In future works it will be necessary to establish a clamp with a standardized pressure higher than that used in this work, to determine the ideal conditions for inducing an effective axonotmesis lesion in the sheep model. Additionally, these results reinforce the idea that evaluation of the regenerative process should always be done in different dimensions, considering not only the signs of histomorphometric regeneration but also the biomechanical and functional recovery, whose results are not always completely corroborative. In the nerves subjected to neurotmesis, there are clear morphological differences between the group that received EtE sutures and NGCs. Comparing the results observed at 12 weeks, it is possible to perceive in both cases axons of smaller diameter compared to the healthy nerve, a situation expected in a short timepoint after injury induction and still under degenerative phenomena. This is more surprising in the neurotmesis + NGC group where regenerating fibers crossed the empty conduit and reached the distal nerve in just 12 weeks, since normally the regeneration takes more time than after EtE suture due to the absence of the extracellular matrix support inside the conduit. These results indicate that good nerve regeneration occurred over the long gap within the NGC, and apparently more effective than that observed by applying the EtE suture, which appears at 12 weeks at an early stage of the regenerative process. 24 weeks after surgery both groups present a more complete regeneration than after 12 weeks, also if they continue to present morphological differences between them. EtE group seems to show a higher density of fibers with a thicker myelin than the NGC group. Additionally, these findings also corroborate the preliminary results observed in both groups in the neurological evaluations performed, and it allows to establish definitively the 24 weeks as the minimum time necessary to evaluate the therapeutic performance after neurotmesis in the sheep model.

Despite the validation of the established surgical method and the findings identified in the functional and stereological evaluation, some doubts remain. Since there was a wide distribution of diameters in the fibers of the healthy nerve, with the presence of small and very large myelinated fibers, it would be important to carry out an immunohistochemical evaluation to distinguish the motor and sensory fibers that constitute this mixed nerve, to fully understand the consequences associated with its injury and also the relationship between the regenerative pattern stereologically observed and the functional recovery. In addition, it is also important in future works to add other methods of assessing functional recovery, such as methods of kinematic and electrophysiological conductivity assessment in the injured nerve, allowing to trace a complete profile of the regenerative behavior of the common peroneal nerve after injury and treatment.

## 4. Materials and Methods

Figure 11 shows the flow chart of the study protocol, including all phases, groups considered, tests performed and numbers of animals per group. All functional assessments and comparisons performed on the animals after nerve injury and on the data obtained in the post-mortem histomorphometric assessment are, at this stage, exclusively exploratory in nature, and as such the levels of significance are not considered.

### 4.1. Animals

All procedures performed on animals were approved by the Organism Responsible for Animal Welfare (ORBEA) of the Abel Salazar Institute for Biomedical Sciences (ICBAS) from the University of Porto (UP) (project 209/2017) and by the Veterinary Authorities of Portugal (DGAV) (project DGAV: 2018-07-11014510). All animal testing procedures were performed in conformity with the Directive 2010/63/EU of the European Parliament and the Portuguese DL 113/2013, and followed the OECD Guidance Document on the Recognition, Assessment and Use of Clinical Signs as Humane Endpoints for Experimental Animals Used in Safety Evaluation (2000). Adequate measures were taken to minimize pain and discomfort, considering humane endpoints for animal suffering and distress.

Ten sheep (*Ovis aries*), merino breed, female gender, 5 to 6 years and 50–60 kg BW were used in this work. All animals were purchased from authorized national producers previously approved by the host institution. Upon arrival, the animals were evaluated, and a prophylactic protocol was instituted (corrective hoof trimming, internal deworming and vaccination against enterotoxaemia). Before being surgically intervened, and regularly throughout the work, the animals were subjected to a general physical examination as well as neurological evaluations. The animals were kept in groups to guarantee the maintenance of their gregarious behavior, were fed with hay and concentrate according to their nutritional needs and had permanent access to fresh water. When subjected to surgery, the animals were pre-anesthetized with xylazine (Rampun^®^, Bayer, Leverkusen, Germany, 0.1 mg/Kg, IM) and butorphanol (Dolorex^®^, Merck Animal Health USA, NJ, USA, 0.05 mg/Kg, IM) and induced with tiletamine and zolazepam (Zoletil^®^, Virbac, Carros, France, 3 mg/Kg, IM). Surgical maintenance was guaranteed with tiletamine and zolazepam (1.5 mg/Kg, IV) and anesthetic recovery was achieved with atipamezole hydrochloride (Antisedan^®^, Zoetis, 0.025 mg/kg IM). After surgery, the animals were treated with anti-inflammatory drugs (meloxicam-Meloxivet^®^, Duprat, Teresina, Brazil, 0.5 mg/Kg, IM, q72h), analgesics (butorphanol, 0.05 mg/Kg, IM) and prophylactic antibiotherapy (ampicillin—Albipen LA^®^, MSD Animal Health, NJ, USA, 15 mg/Kg, q48 h) during one week. After the corresponding study period, the animals were sedated with the same protocol applied pre-surgically, and then euthanized using an overdose of sodium pentobarbital (Eutasil^®^, Ceva Animal Health Solutions, Libourne, France, 100 mg/Kg IV).

### 4.2. Surgery

#### 4.2.1. Surgical Preparation

After pre-anesthetic induction, the animals were placed in a right lateral recumbency over the surgery table, followed by the preparation of the surgical field (trichotomy of the proximal region of the hind limb, cleaning and asepsis and placement of a surgical drapes). Then, the local block of the common peroneal nerve was performed, with administration of approximately 2–5 mL of lidocaine (Anestesin^®^, Medinfar, Lisbon, Portugal, 1.7 mg/Kg), in the lateral surface of the hind limb, in the region where the nerve runs obliquely, about 2.5 cm below the tibial condyle. 

#### 4.2.2. Surgical Access

Once the local nerve block was achieved, the nerve could be surgically accessed. Three different surgical approaches were initially considered: (1) exposure of the common peroneal nerve through a lateral incision along the thigh (Figure 12); (2) exposure of the common peroneal nerve more distally, at the location where the nerve branches into the superficial and deep common peroneal nerves, between the peroneus longus and lateral digital extensor muscles; (3) caudal access, through the separation between the semitendinosus and semimembranosus muscles.

Due to the ease of implementation and the few associated risks, as well as due to the disadvantages associated with the technique 2 and 3, it was decided to only apply technique 1 in the subsequent phases. In this technique, the incision should start at the level of the patella and advance along the tibia, in plantar position, ending 2 cm distally to the crest of the tibia. After the skin incision, it is possible to immediately observe the insertion of the biceps femoris muscle and the common peroneal nerve appearing under it. Subcutaneous debridement should be carried out in such a way as to individualize the common peroneal nerve from neighboring tissues. The ventral-cranial disinsertion of the biceps femoris muscle allows greater exposure of the nerve, and the extension of the disinsertion must be adapted to the anatomical characteristics of each animal. 

#### 4.2.3. Nerve Injury and Therapeutic Options

Once the common peroneal nerve was individualized, the animals were subjected to different types of nerve damage. Table 3 indicates the groups stablished, types of injuries induced, therapeutic approaches and study times considered. The analogue contralateral nerve was considered as healthy control. 

In the neurotmesis lesion, after immobilizing the nerve with tweezers, in a non-traumatic way, a complete transection of the nerve was performed using a scalpel (Figure 13a). The transection must be performed with a sharp scalpel blade, and with a single movement, to allow a clean cut and to avoid irregular nervous tops. After inducing the neurotmesis lesion, two therapeutic approaches were considered: (1) End-to-end (EtE) tension-free suture, in which, after removing irregularities and portions of excessively damaged nervous tissue, the nerves were coaptized to maintain the original anatomical orientation and leaving a minimum gap between the nerve tops, being kept in this position through the application of individual epineural sutures that guarantee physiological alignment and avoid rotations. Microsutures were performed with 7/0 monofilament polyglycolic acid material (Safil^®^) (Figure 13b); (2) application of a NGC, 3 cm long and 3.0 mm in diameter (Reaxon^®^ Nerve Guide, Medovent, Mainz, Germany), in which the nerve tops were introduced 3 mm, leaving a gap of approximately 24 mm between them. To guarantee the fixation of the nerves aligned and in an anatomical position, the epineurium was sutured to the NGC with 7/0 monofilament polyglycolic acid material (Figure 13c). For the axonotmesis lesion, a non-serrated clamp was used (Figure 14a). Initially, a technique similar to that used in a rat model by Varejão et al. [65] was applied. However, the absence of symptoms in the post-surgical period demonstrated the ineffectiveness of this clamp to induce a crush injury in the sheep. The clamp was then adapted, receiving a spring capable of inducing a force of 80 N and an adapter to increase the crushing surface to 5 mm in length. The sheep’s common peroneal nerve has an initial diameter of approximately 3 mm, ending, after crushing, with a final diameter of approximately 4 mm. Thus, the final pressure, exerted for 1 min, was *p* = 4 MPa. After removing the clamp, the flattened region corresponding to the crushing area was observed (Figure 14b). 

After lesion induction and therapeutic applications, the subcutaneous and cutaneous layers of the surgical access were closed with simple interrupted sutures with non-absorbable 4/0 material. To avoid abrasion lesions of the limb ends during gait, due to expected functional impairments, a padded bandaged was applied. For the neurotmesis lesion, recovery times of 12 and 24 weeks were considered for each type of therapeutic approach. For axonotmesis injuries, 12 weeks were considered. 

### 4.3. Neurological Evaluation

After induction of PNI, the animals were subjected to a neurological examination adapted to the common Peroneal Nerve and involving an assessment of mental status, movements and posture, postural reactions, and spinal reflexes. The components considered during the neurological exam are described in Table 4. The animals were evaluated preoperatively to establish baseline values, one week after the surgical injury, and thereafter every two weeks until the end of the study period for each type of injury. The combination and sequence of tests and assessments considered in this adapted neurological exam was created by the authors.

The assessment of the mental status involved determining the animal’s ability to perceive its surrounding environment, the presence of the operator and also its alertness. The animal’s posture was evaluated in a stationary position, both at the level of the digits and the hook, observed in lateral and posterior views. The free movements were assessed in a wide space, without obstacles, inducing the animal to perform movements in a straight line, with circular movements and with changes in speed and direction. During the movements, the agility of the animal and possible manifestations of pain were evaluated. Postural reactions were assessed on both the injured and healthy limbs for comparison. The proprioceptive assessment was performed using the static and dynamic proprioceptive positioning tests, in both cases quantifying the time, in seconds, that the animal took to reposition the limb to its physiological position. Finally, the evaluation of spinal reflexes was performed through the withdrawal reflex, with the animal in lateral recumbence, with the limb to be evaluated facing upwards, using a hoof forceps to stimulate the skin dorsal to the lateral digits and the digits themselves (Figure 2).

### 4.4. Nerve Morphological and Stereological Analysis

In each group, after the considered time periods, euthanasia was performed as described, and the injured nerves were collected. Contralateral healthy nerves were also collected as healthy controls. In both cases, the nerves were fixed and adequately prepared to be analyzed histomorphometrically by light microscopic examination.

After transcutaneous access and careful nerve exposure, fixation was performed with a few drops of fixation solution consisting of 2.5% purified glutaraldehyde and 0.5% saccharose in 0.1 M Sorensen phosphate buffer at pH 7.4, kept at 4 °C, stiffening the nerve and facilitating its collection. Then the desired nerve segment was collected and dipped, properly oriented and stowed, in the same fixation solution for 5 min. Finally, the collected segments were kept immersed in the fixing solution for 6–8 h, after which they were washed abundantly in a washing solution consisting of 1.5% saccharose in 0.1 M Sorensen phosphate buffer at pH 7.4, being kept immersed in this solution until analysis. Histomorphometric analysis was performed according to a protocol previously used [68], and the parameters total number of fibers (N), fiber density (N/mm^2^), axon diameter (d, µ), fiber diameter (D, µ), myelin thickness (M, µ) and cross-sectional area (mm^2^) were considered, in addition to the ratios d/D (g-ratio), M/d, D/d. 

## 5. Conclusions and Further Directions

This article describes an easily applicable and straightforward pre-clinical injury protocol for the common peroneal nerve in the sheep model, including the surgical approach for the induction of crush injuries and neurotmesis, as well as the methodology for applying therapeutic tactics. A detailed description of the anatomy and functionality of the sheep’s hind limb nerves was included, allowing a complete understand on the variations associated to the lesions of the selected nerve in apposition to other neighboring peripheral nerves. In addition, a neurological exam protocol to monitor the functional evolution over the regeneration period was stablished, followed by a neurological evaluation in the different animals intervened to confirm the effectiveness of the selected exam in identifying functional variations over time. Finally, a stereological and histomorphological evaluation of the intervened and healthy nerves to register control values for the sheep model and identify preliminary morphological differences between the therapeutic options was performed. It was also possible to define a lateral incision along the thigh as the ideal way to access and induce lesions in the common peroneal nerve, besides stablishes the study times of 12 weeks for axonotmesis injuries and 24 weeks for neurotmesis injuries as those necessary for this specie and type of studies extended in time. Although the data is insufficient to draw deeper conclusions, the preliminary results observed show a tendency to better regenerative outcomes in the NGC group. However, this result is merely indicative and no conclusion can be supported, as the sample number is too small. In the case of axonotmesis injuries, despite post-injury observation of symptomatology compatible with peroneal nerve lesion, an incomplete injury observed morphologically demonstrated the need to use higher pressure/crushing times to guarantee an effective crush injury in future essays. This model may be useful in upcoming works to evaluate therapeutic options and repair strategies and its real effectiveness in promoting nerve regeneration after PNI. Finally, the article aims to encourage other researchers to use sheep as a relevant pre-clinical model of PNI and the protocol created can now be used in the development of new strategies relevant to clinical treatment in human and veterinary medicine.

## Figures and Tables

**Figure 1 ijms-22-01401-f001:**
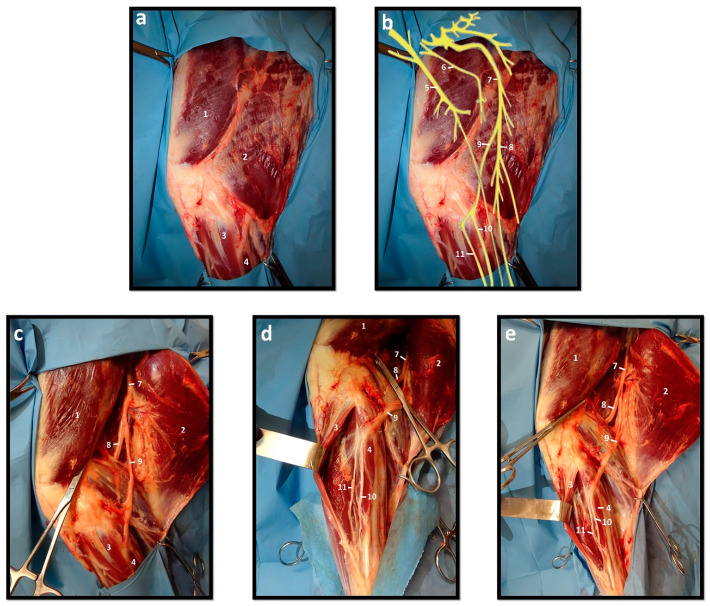
Anatomical and topographic distribution of sheep’s hind limb nerves. (**a**) Muscles of the sheep’s hind limb—lateral view; (**b**) Schematic representation of the topographic distribution of the main nerves in the sheep’s hind limb; (**c**) Muscles and nerves of the sheep’s hind limb—deep exposure of the proximal region; (**d**) Muscles and nerves of the sheep’s hind limb—deep exposure of the distal region; (**e**) Muscles and nerves of the sheep’s hind limb—deep exposure. (1. M. vastus lateralis; 2. M. biceps femoris; 3. M. extensor digitalis lateralis; 4. M. peroneus longus; 5. Femoral n.; 6. Obturator n.; 7. Sciatic n.; 8. Tibial n.; 9. Common peroneal n.; 10. Superficial peroneal n.; 11. Deep peroneal n.).

**Figure 2 ijms-22-01401-f002:**
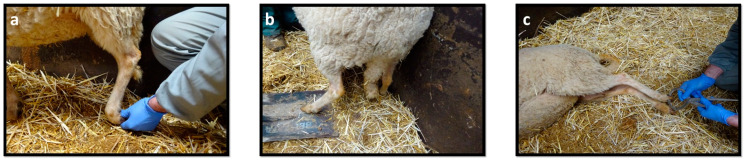
Components of the neurological exam applied to sheep: (**a**) Static repositioning; (**b**) Dynamic repositioning; (**c**) Withdrawal reflex.

**Figure 3 ijms-22-01401-f003:**
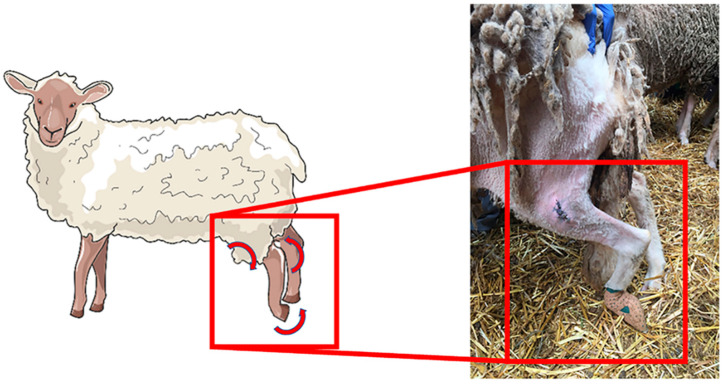
Schematic representation (**left**) and sheep after 1 week common peroneal nerve injury (**right**): Overextension of the hock and overflexion of the distal joints, with the limb resting on the dorsal surface of the flexed digits.

**Figure 4 ijms-22-01401-f004:**
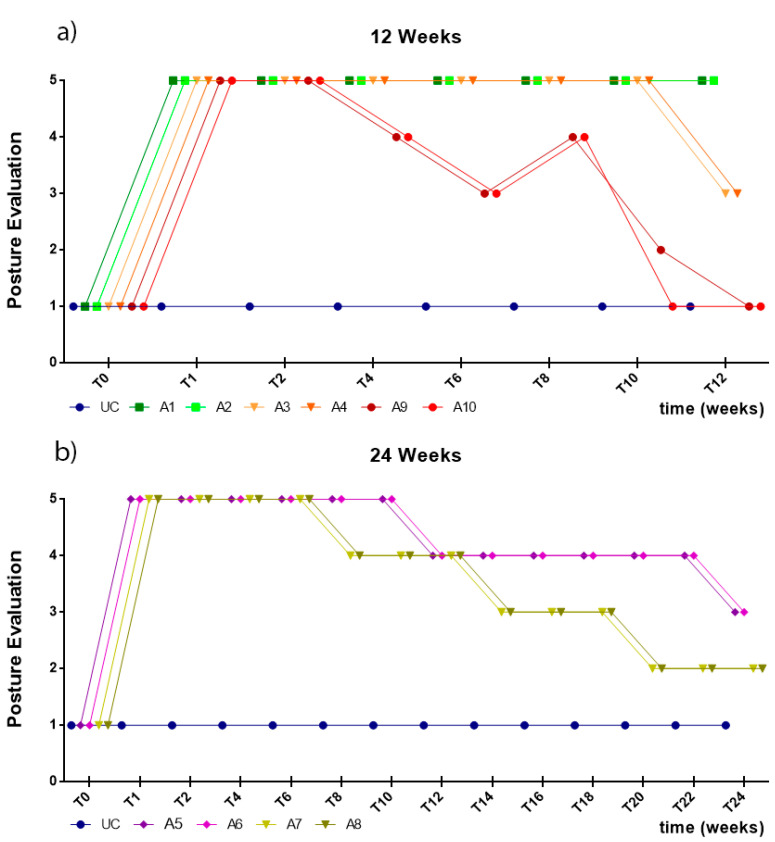
Results of posture evaluation performed on all animals over the study periods established for each type of injury. UC: Uninjured Control; A1 and A2: Neurotmesis + EtE 12 weeks; A3 and A4: Neurotmesis + NGC 12 weeks; A5 and A6: Neurotmesis + EtE 24 weeks; A7 and A8: Neurotmesis + NGC 24 weeks; A9 and A10: Axonotmesis 12 weeks. Classification key—1: Digits and hock in physiological position, no postural changes; 2: Mild flexion of digits and/or extension of the hock; 3: Moderated flexion of digits and/or extension of the hock; 4: Pronounced flexion of digits and extension of the hock; 5: Severe flexion of digits and extension of the hock.

**Figure 5 ijms-22-01401-f005:**
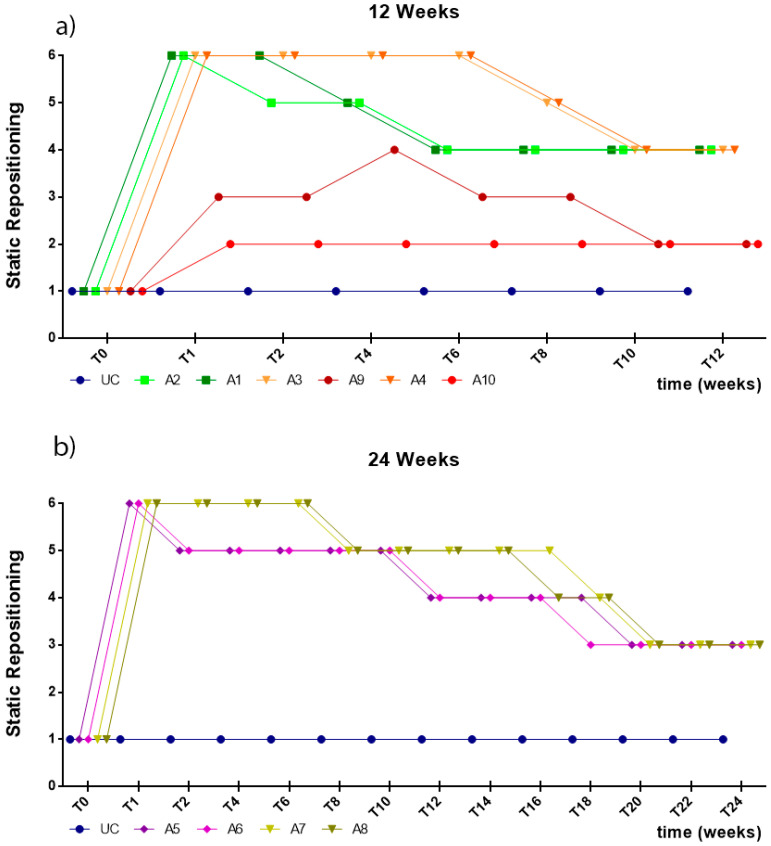
Results of proprioceptive assessment using the static repositioning test, performed on all animals over the study periods established for each type of injury. UC: Uninjured Control; A1 and A2: Neurotmesis + EtE 12 weeks; A3 and A4: Neurotmesis + NGC 12 weeks; A5 and A6: Neurotmesis + EtE 24 weeks; A7 and A8: Neurotmesis + NGC 24 weeks; A9 and A10: Axonotmesis 12 weeks. Classification key—1: <3 s; 2: 3–5 s; 3: 5–10 s; 4: 10–15 s; 5: 15–20 s; 6: >20 s.

**Figure 6 ijms-22-01401-f006:**
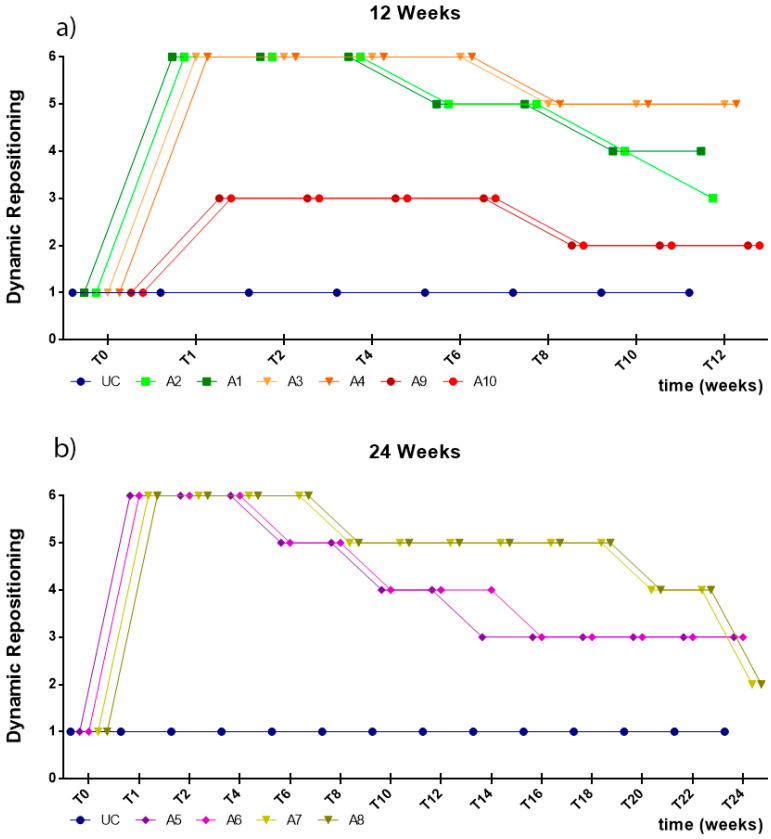
Results of proprioceptive assessment using the dynamic repositioning test, performed on all animals over the study periods established for each type of injury. UC: Uninjured Control; A1 and A2: Neurotmesis + EtE 12 weeks; A3 and A4: Neurotmesis + NGC 12 weeks; A5 and A6: Neurotmesis + EtE 24 weeks; A7 and A8: Neurotmesis + NGC 24 weeks; A9 and A10: Axonotmesis 12 weeks. Classification key—1: <3 s; 2: 3–5 s; 3: 5–10 s; 4: 10–15 s; 5: 15–20 s; 6: >20 s.

**Figure 7 ijms-22-01401-f007:**
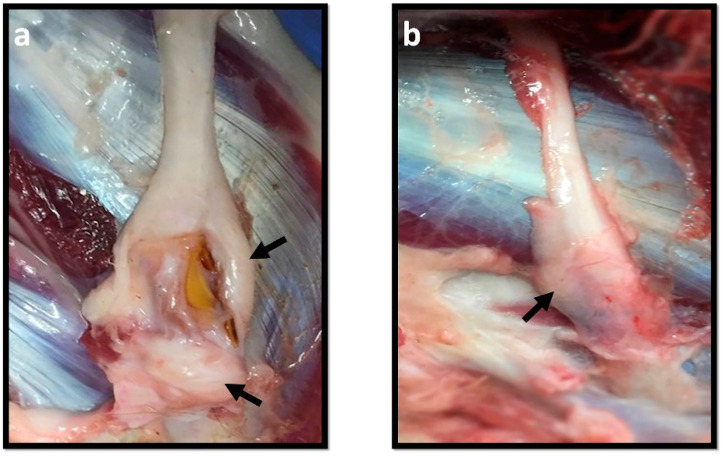
Macroscopic aspect of the nerves during collection. The black arrows show the presence of fibrous tissue around the nerve. (**a**) Nerve that received NGC; (**b**) Nerve that received EtE.

**Figure 8 ijms-22-01401-f008:**
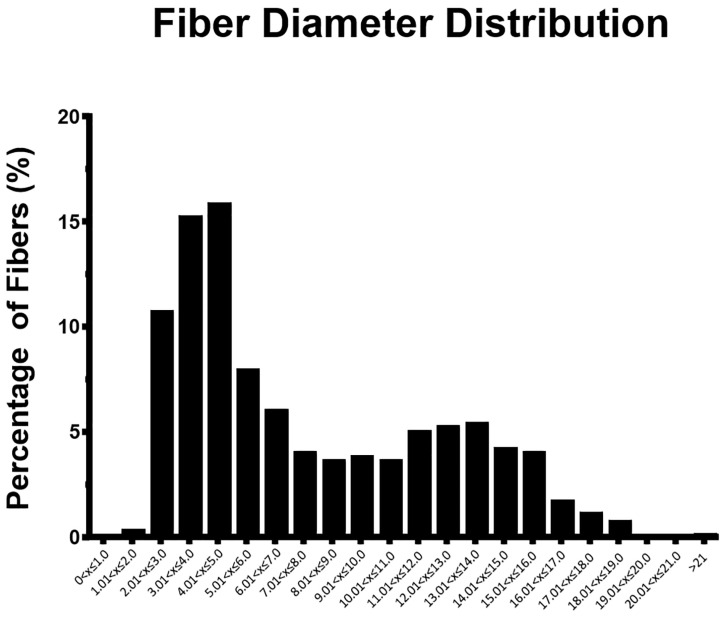
Histogram showing the distribution of the diameters of the healthy peroneal common nerve fibers.

**Figure 9 ijms-22-01401-f009:**
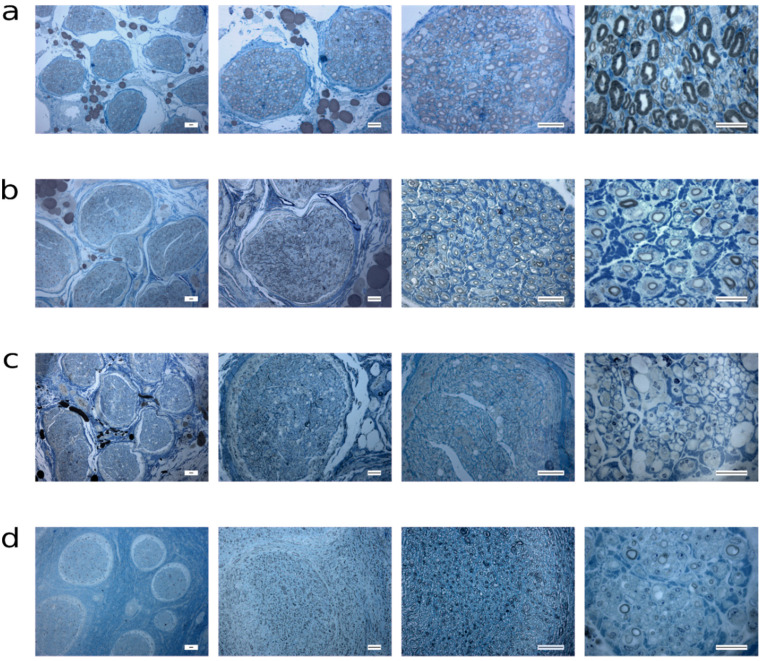
Light micrographs of Toluidine blue-stained common peroneal semi-thin sections for the different groups at 12 weeks. (**a**) Healthy nerve; (**b**) Axonotmesis; (**c**) Neurotmesis + EtE; (**d**) Neurotmesis + NGC. Scale bars: first column = 100 μm; second and third columns = 40 μm; fourth column = 20 μm.

**Figure 10 ijms-22-01401-f010:**
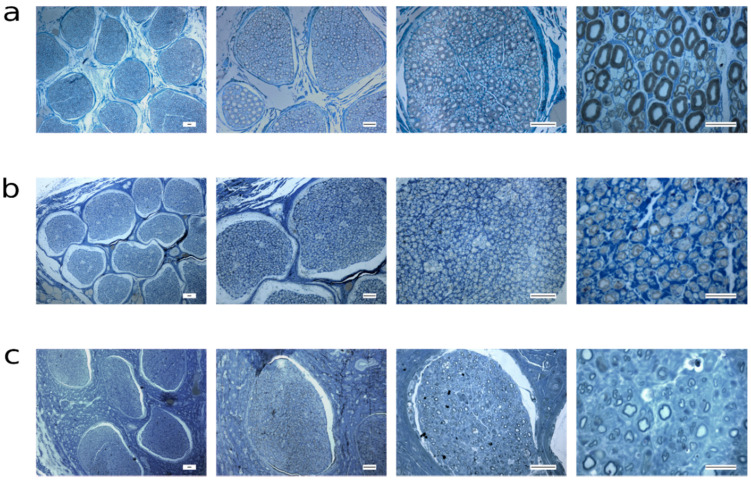
Light micrographs of Toluidine blue-stained common peroneal semi-thin sections for the different groups at 24 weeks. (**a**) Healthy nerve; (**b**) Neurotmesis + EtE; (**c**) Neurotmesis + NGC. Scale bars: First column = 100 μm; second and third columns = 40 μm; fourth column = 20 μm.

**Figure 11 ijms-22-01401-f011:**
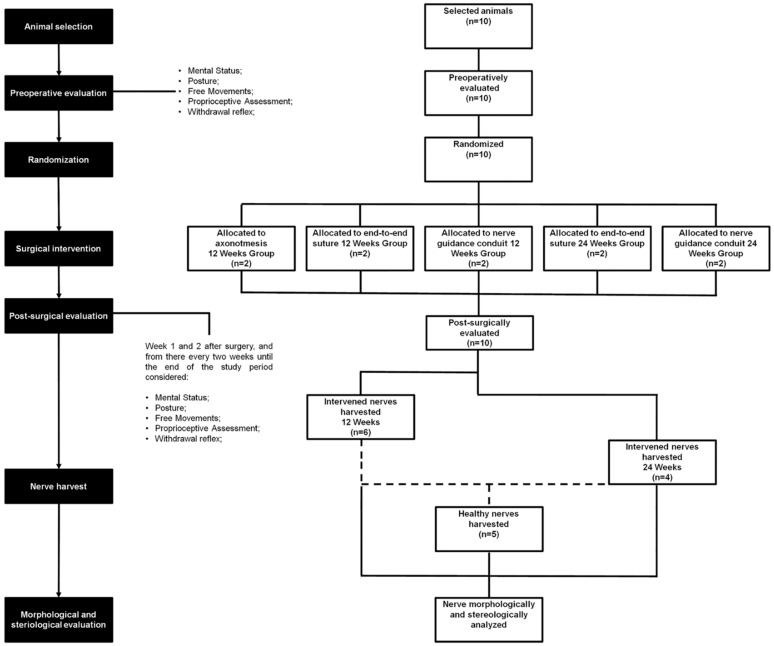
Flow chart of the study protocol.

**Figure 12 ijms-22-01401-f012:**
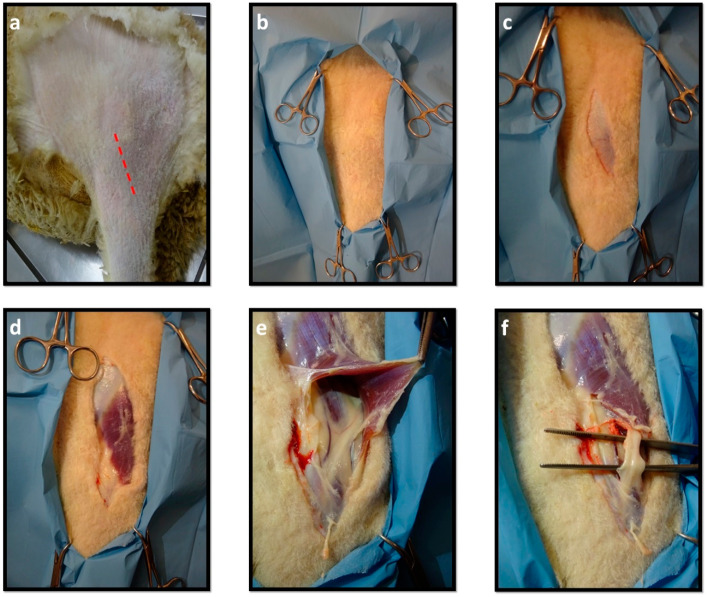
Sequence of steps for access and exposure of the common peroneal nerve: (**a**) Trichotomy of the hind limb´s proximal region; (**b**) Delimitation of the region to intervene with surgical drapes; (**c**) Skin incision beginning at the level of the patella, advancing along the tibia, in plantar position, ending 2 cm distally to the crest of the tibia; (**d**) Debridement of subcutaneous tissue with exposure of the cranial portion of the M. biceps femoris; (**e**) Disinsertion of the M. biceps femoris for better exposure of the common peroneal nerve; (**f**) Individualization of the common peroneal nerve.

**Figure 13 ijms-22-01401-f013:**
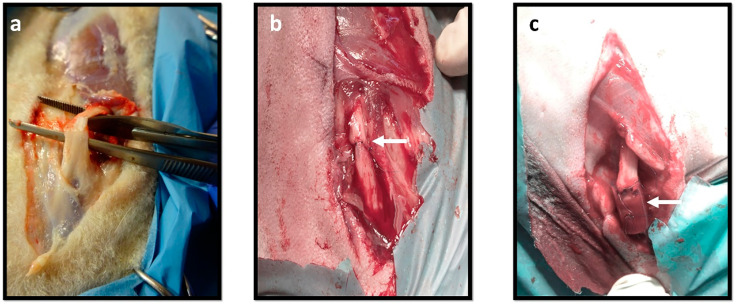
Neurotmesis Injury: (**a**) Induction of neurotmesis injury with a scalpel blade; (**b**) Nerve with neurotmesis lesion and with EtE sutures; (**c**) Nerve with neurotmesis lesion and with a NGC.

**Figure 14 ijms-22-01401-f014:**
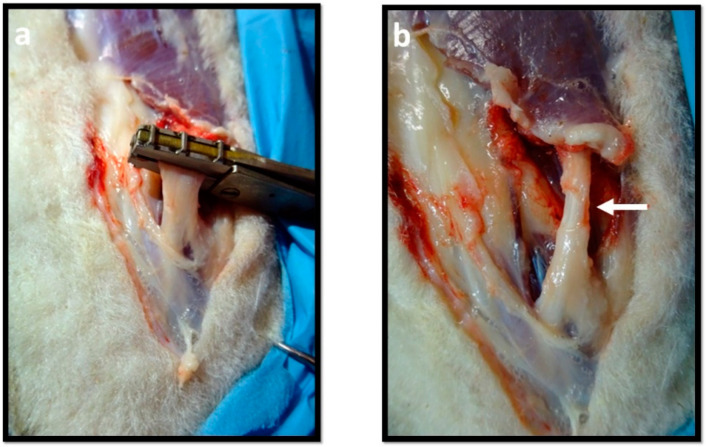
Axonotmesis Injury: (**a**) Induction of an axonotmesis injury with a non-serrated clamp; (**b**) Flattened region observable in the nerve after the crushing injury.

**Table 1 ijms-22-01401-t001:** Results of the withdrawal reflex assessment, performed on all animals over the study periods established for each type of injury. UC: Uninjured Control. Classification key: +: Absent reflex; ++: Reflex present but delayed; +++: Reflex present.

Withdrawal Reflex		Time													
		T0	T1	T2	T4	T6	T8	T10	T12	T14	T16	T18	T20	T22	T24
Group 1: UC		+++	+++	+++	+++	+++	+++	+++	+++	+++	+++	+++	+++	+++	+++
Group 2: Neurotmesis + End to End, 12 W	A1	+++	+	+	+	++	++	++	+++						
	A2	+++	+	+	+	++	++	+++	+++						
Group 3: Neurotmesis + Nerve Guidance Conduit, 12 W	A3	+++	+	+	+	+	+	+	++						
	A4	+++	+	+	+	+	++	++	++						
Group 4: Neurotmesis + End to End, 24 W	A5	+++	+	+	+	++	++	++	+++	+++	+++	+++	+++	+++	+++
	A6	+++	+	+	+	++	++	++	++	++	+++	+++	+++	+++	+++
Group 5: Neurotmesis + Nerve Guidance Conduit, 24 W	A7	+++	+	+	+	+	++	++	++	++	++	+++	+++	+++	+++
	A8	+++	+	+	+	+	++	++	++	++	+++	+++	+++	+++	+++
Group 6: Axonotmesis, 12 W	A9	+++	+	+	++	++	++	++	+++						
	A10	+++	+	+	+	++	++	++	++						

**Table 2 ijms-22-01401-t002:** Stereological quantitative assessment. The different parameters considered were evaluated on sheep´s healthy common peroneal nerves. Results are presented as mean and SD (n = number of animals per group).

Stereological Quantitative Assessment		Density	Total Number	Axon Diameter (d)	Fiber Diameter (D)	Myelin Thickness (M)	M/d	D/d	d/D (g-Ratio)	Cross-Sectional Area (mm^2^)
Healthy nerves (n = 5)	Mean	11.969	21.154	4.51	7.73	1.61	0.36	1.72	0.60	1.87
	SD	3193	3937	0.83	1.60	0.41	0.07	0.14	0.04	0.53

**Table 3 ijms-22-01401-t003:** Established therapeutic groups.

Injury	Treatment	Time Point (Weeks)	Animals
Neurotmesis	End-to-end suture	12	2
Neurotmesis	Nerve guidance conduit	12	2
Neurotmesis	End-to-end suture	24	2
Neurotmesis	Nerve guidance conduit	24	2
Axonotmesis	Without treatment	12	2

**Table 4 ijms-22-01401-t004:** Parameters considered during the neurological exam applied to sheep. Classification key: Animal—Animal identification; Mental Status—1: Alert and responsive; 2: Obtunded, 3: Stuporous; 4: Semicomatose; 5: Comatose and unresponsive; Posture Evaluation—1: Digits and hock in physiological position, no postural changes; 2: Mild flexion of digits and/or extension of the hock; 3: Moderated flexion of digits and/or extension of the hock; 4: Pronounced flexion of digits and extension of the hock; 5: Severe flexion of digits and extension of the hock; Movements Evaluation—1: Free and voluntary movements, absence of discomfort and/or pain; 2: Voluntary movements, manifestation of discomfort; 3: Voluntary movements, manifestation of discomfort and/or pain; 4: Conditioned voluntary movements, manifestation of discomfort and/or pain; 5: Absence of voluntary movements, manifestation of discomfort and/or pain; Postural Reactions (time for limb repositioning)—1: <3 s; 2: 3–5 s; 3: 5–10 s; 4: 10–15 s; 5: 15–20 s; 6: >20 s. Spinal Reflexes +: Absent reflex; ++: Reflex present but delayed; +++: Reflex present.

Animal	Mental Status	Movement and Posture		Postural Reactions				Spinal Reflexes	
		Stationary Position (Posture Evaluation)	Free Movements	Health Limb (Seconds)		Injured Limb (Seconds)			
				Static Repositioning	Dynamic Repositioning	Static Repositioning	Dynamic Repositioning	Parameters	Withdrawal Reflex
								Proximal skin	
								Distal skin	
								Digits	

## Data Availability

The data that support the findings of this study are available from the corresponding author on request.

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
