# Peer review of "Establishment of a Sheep Model for Hind Limb Peripheral Nerve Injury: Common Peroneal Nerve"

_ijms, 2021, doi:10.3390/ijms22031401_

Round 1

Reviewer 1 Report

The authors present a well-structured article about the assessment and validation of a sheep model of common peroneal nerve injury. Therefore, the article includes a very detailed (sometimes, too detailed) description of the anatomy and functionality of the peripheral nerves of sheep’s hind limbs, a defined surgical protocol, and different therapeutic strategies.

Whereas the authors are able to convince me about the role and validity of such a model, I have different issues that should be addressed.

The first one is the article's length: sometimes, it is really too long, i.e. about the anatomical description or in methods (i.e. paragraph 4.3 - Neurological Evaluation - could its length be reduced?

I suggest combining Figure 2 (Schematic representation of a sheep with common peroneal nerve injury) with Figure 7 (Posture of a sheep with a 1-week common peroneal nerve injury) to obtain one single figure, placed in the place of fig 7.

Moreover, I think that figure 6 (Components of the neurological exam applied to sheep) should be placed within paragraph 1.4 (Neurological exam directed to the hind limb) to better explicated the paragraph itself

Reviewer 2 Report

1) The manuscript is too long and the structure needs improvement. Especially the introduction needs to go more to the point. In the current form readability is limited.

2) The Abstract needs structure and statistics.

3) provide a flow Chart of the study protocol and the tests that were performed with numbers of specific groups

4) Overall numbers of N=2 in Group 2-6 doesn't allow for sound statistical analysis. Explainability is limited.

Round 2

Reviewer 1 Report

I think that now the manuscript is more suitable for publication

Author Response

The authors would like to thank once again for the revision and additional comments, which significantly improved the manuscript.

Reviewer 2 Report

Some of my comments were adequately answered. Especially the flowchart helps to understand the study design. My main point, low numbers (n=2) was not addressed. 

Author Response

Answer to Reviewer 2

Dear Reviewer 2:

Thank you very much for the feedback on this review phase, and also for the suggestions made, which received the best attention from us. We effectively understand the question that continues to be raised by the reviewer regarding the sample number (n) that does not allow us to infer broader conclusions or declare statistical relevance. At the same time, as mentioned, we cannot increase the n at this stage. Aware of this limitation, it was from the beginning our intention that the functional evaluations performed had the sole objective of understanding whether the applied injury models allow the subsequent performance of a functional assessment, allowing to identify nervous deficits and their progression over time and to establish preliminary reference data for future works. That is, regarding the values obtained and compared in the functional evaluation and in the post-mortem histomorphometric evaluation, these are exclusively preliminary and exploratory, allowing to establish the ovine model described as valid and to obtain guiding values that can be used as a reference for future works on peripheral nerve injury. At any point in the text was any statistical analysis or relevance established or referred to. In addition to the various passages in the text where this exploratory approach was explained, as mentioned in our previous answer, the authors decided to introduce the following paragraph in the "Materials and Methods" section to reinforce the information again (Line 670-673, highlighted in yellow in the main document):

“All functional assessments and comparisons performed on the animals after nerve injury and on the data obtained in the post-mortem histomorphometric assessment are, at this stage, exclusively exploratory in nature, and as such the levels of significance are not considered.”:

Additionally, and aware of the improvement that could result from it, the following changes were also introduced:

Tables 1, 2 and 3 were replaced by graphs with the same information, visually improving the exposure of the data, showing the data of each animal individually and not using mean and SD values. The new figures were named Figures 4, 5 and 6, which caused changes in the numbering of the remaining figures and in the tables. The changes were introduced in the final document and the new figure caption appears highlighted in yellow.

Figure 4 - Results of posture evaluation performed on all animals over the study periods established for each type of injury. UC: Uninjured Control; A1 and A2: Neurotmesis+EtE 12 weeks; A3 and A4: Neurotmesis+NGC 12 weeks; A5 and A6: Neurotmesis+EtE 24 weeks; A7 and A8: Neurotmesis+NGC 24 weeks; A9 and A10: Axonotmesis 12 weeks. Classification key1: Digits and hock in physiological position, no postural changes; 2: mild flexion of digits and/or extension of the hock; 3: Moderated flexion of digits and/or extension of the hock; 4: Pronounced flexion of digits and extension of the hock; 5: Severe flexion of digits and extension of the hock.

Figure 5 - Results of proprioceptive assessment using the static repositioning test, performed on all animals over the study periods established for each type of injury. UC: Uninjured Control; A1 and A2: Neurotmesis+EtE 12 weeks; A3 and A4: Neurotmesis+NGC 12 weeks; A5 and A6: Neurotmesis+EtE 24 weeks; A7 and A8: Neurotmesis+NGC 24 weeks; A9 and A10: Axonotmesis 12 weeks. Classification key - 1: <3s; 2: 3-5s; 3: 5-10s; 4: 10-15s; 5: 15-20s; 6: >20s.

Figure 6 - Results of proprioceptive assessment using the dynamic repositioning test, performed on all animals over the study periods established for each type of injury. UC: Uninjured Control; A1 and A2: Neurotmesis+EtE 12 weeks; A3 and A4: Neurotmesis+NGC 12 weeks; A5 and A6: Neurotmesis+EtE 24 weeks; A7 and A8: Neurotmesis+NGC 24 weeks; A9 and A10: Axonotmesis 12 weeks. Classification key - 1: <3s; 2: 3-5s; 3: 5-10s; 4: 10-15s; 5: 15-20s; 6: >20s.

  • Table 4 has been modified, and instead of numerical values the assessment of the withdrawal reflex is now done by rating +/++/+++. In this way the information becomes easier to interpret and it is easier for the reader to understand the functional variation of animals and groups over time. Due to the changes inserted in the previous point, the table was renamed as Table 1, with changes in the numbering of all other tables. The changes were introduced in the final document and the new table caption appears highlighted in yellow.

Table 1. - Results of the withdrawal reflex assessment, performed on all animals over the study periods established for each type of injury. UC: Uninjured Control. Classification key - +: Absent reflex; ++: Reflex present but delayed; +++: Reflex present.
